# Calcitonin Gene-Related Peptide Systemic Effects: Embracing the Complexity of Its Biological Roles—A Narrative Review

**DOI:** 10.3390/ijms241813979

**Published:** 2023-09-12

**Authors:** Adriano Bonura, Nicoletta Brunelli, Marilena Marcosano, Gianmarco Iaccarino, Luisa Fofi, Fabrizio Vernieri, Claudia Altamura

**Affiliations:** 1Instituite of Neurology, Fondazione Policlinico Universitario Campus Bio-Medico, Via Alvaro del Portillo, 200, 00128 Roma, Italy; adriano.bonura@unicampus.it (A.B.); n.brunelli@policlinicocampus.it (N.B.); marilena.marcosano@unicampus.it (M.M.); l.fofi@policlinicocampus.it (L.F.); f.vernieri@policlinicocampus.it (F.V.); 2Unit of Headache and Neurosonology, Department of Medicine and Surgery, Università Campus Bio-Medico di Roma, 00128 Roma, Italy

**Keywords:** calcitonin gene-related peptide (CGRP), migraine, comorbidity

## Abstract

The calcitonin gene-related peptide (CGRP) is a neuropeptide widely distributed throughout the human body. While primarily recognized as a nociceptive mediator, CGRP antagonists are currently utilized for migraine treatment. However, its role extends far beyond this, acting as a regulator of numerous biological processes. Indeed, CGRP plays a crucial role in vasodilation, inflammation, intestinal motility, and apoptosis. In this review, we explore the non-nociceptive effects of CGRP in various body systems, revealing actions that can be contradictory at times. In the cardiovascular system, it functions as a potent vasodilator, yet its antagonists do not induce arterial hypertension, suggesting concurrent modulation by other molecules. As an immunomodulator, CGRP exhibits intriguing complexity, displaying both anti-inflammatory and pro-inflammatory effects. Furthermore, CGRP appears to be involved in obesity development while paradoxically reducing appetite. A thorough investigation of CGRP’s biological effects is crucial for anticipating potential side effects associated with its antagonists’ use and for developing novel therapies in other medical fields. In summary, CGRP represents a neuropeptide with a complex systemic impact, extending well beyond nociception, thus offering new perspectives in medical research and therapeutics

## 1. Introduction

Calcitonin Gene-Related Peptide (CGRP) is a neuropeptide consisting of 37 amino acids [1]. It was discovered approximately 40 years ago as a product of alternative splicing of the calcitonin gene mRNA and isolated from the thyroid of patients with thyroid carcinoma [2]. It belongs to the family of adrenomedullins, calcitonin, and amylin and is primarily located in the central nervous system, particularly in the hypothalamus, olfactory system, and gustatory system [1]. In the peripheral nervous system, it is found in type C and Aδ sensory fibers that play roles both as afferents (nociceptive) and efferents and are present near blood vessels in all organs of the body [3,4].

CGRP is primarily known for its role in pain perception processes associated with migraine [5,6,7]. It is released by the activated trigeminovascular system and is responsible for both migraine pain perception and the direct induction of migraine attacks [5,6,7]. This has generated strong pharmacological interest, and in the 1990s, it was discovered that triptans, drugs already used for decades in migraine, inhibit CGRP release [8]. Based on this evidence, drugs with a direct action on CGRP have been subsequently developed, such as monoclonal antibodies and gepants targeting CGRP and its receptor [1,9].

In addition to its role as a pain mediator, CGRP is the most potent vasodilator discovered to date [1]. Even the injection of small amounts of CGRP at the cutaneous level causes massive vasodilation of arterioles and an increase in blood flow in the inoculated area [10]. These effects indicate an important role of CGRP in the cardiovascular system. Furthermore, the ubiquitous distribution of CGRP in the body stimulates interest in the systemic effects that this molecule could have, both physiologically and pathologically. A comprehensive analysis of CGRP effects is also important, considering the increasing use of its antagonists in migraines to understand and predict the short-term and long-term effects of these drugs [11].

The purpose of this review is therefore to summarize the current understanding of the role of CGRP in the physiology and pathophysiology in different body districts. The synthesis and release of CGRP, as well as the biological effect of binding to its receptors, will be analyzed. Subsequently, the non-nociceptive and non-neurological effects of CGRP on different organs and systems will be examined.

This narrative review included a search of literature through PubMed of articles in English. We searched for studies that included “CGRP”, “Calcitonin gene related peptide”, “cardiovascular”, “heart”, “vessels”, “gastrointestinal”, “skin”, “systemic effects”, “immunology”, and “musculoskeletal”. We screened abstracts and then extracted information relevant to this review.

## 2. CGRP Biology

### 2.1. CGRP Isoforms

CGRP exists in two main forms, α and β, synthesized from two distinct genes located at different sites on chromosome 11 in humans [12,13]. The CALC I gene is part of a gene family that includes adrenomedullin (1 and 2), and amylin, and is responsible for the production of calcitonin or the alpha isoform of CGRP if alternative splicing occurs [1]. The CALC II gene determines only the formation of β-CGRP [1]. CGRP α and CGRP β share approximately 90% homology and exhibit similar biological activities [1].

### 2.2. Regulation of CGRP Synthesis and Release

The regulation of CGRP synthesis is still poorly understood [1]. It appears to be enhanced in situations of damage to the nerve endings of C or Aδ sensory fibers or in the presence of inflammation in tissues adjacent to the fibers [14]. Nerve growth factor (NGF) is involved in the growth and function of sensory nerves and appears to play an important role in increasing CGRP production in dorsal root ganglia [15]. Furthermore, elevated levels of NGF and CGRP are observed in patients with migraines [16]. Other factors, such as brain-derived neurotrophic factor (BDNF), may influence CGRP release and activity [17]. The Calcitonin Gene-Related Peptide (CGRP) is co-localized in pain nerve terminals along with Substance P (SP), which exhibits similar biological effects. SP acts as a cutaneous vasodilator and increases microvascular permeability, with an earlier release and less potent and enduring effects compared to CGRP [18]. SP serves as a key mediator of pain perception in pain modulation and sensation. Additionally, it is also involved in the regulation of inflammation, nausea, and vomiting, and the modulation of behavioral disorders [18].

After synthesis, CGRP is stored in dense vesicles within sensory nerve endings and is released through calcium-dependent exocytosis [19,20]. The release of CGRP occurs following the activation of TRPV1 and TRPA1 receptors [21,22]. TRPV1 can be activated by painful thermal stimuli, pH reduction, and inflammation [22].

Additionally, CGRP release also appears to occur in response to vasoconstrictors such as angiotensin II and noradrenaline, which activate α2 adrenergic receptors with NGF-induced CGRP release [23].

### 2.3. CGRP Receptor

The CGRP receptor consists of two subunits: the calcitonin-like receptor (CLR) and the receptor activity-modifying protein (RAMP). CLR belongs to the class B of G protein-coupled receptors (GPCR), which also includes receptors for calcitonin, vasoactive intestinal peptide (VIP), pituitary adenylate cyclase-activating polypeptide (PACAP), and parathyroid hormone (PTH) [1]. The RAMP protein family consists of three members: RAMP1, RAMP2, and RAMP3 [1].

The co-expression of CLR and RAMP1 forms a high-affinity receptor for CGRP (CGRP1 receptor), while the dimerization of CLR and RAMP2 creates a receptor highly responsive to adrenomedullin (AM1 receptor) [1]. The RAMP3 receptor confers a second receptor for adrenomedullin (AM2 receptor) with some selectivity for CGRP [24,25]. Indeed, although the true receptor for CGRP is considered to be formed by the CLR/RAMP1 complex, in vitro studies have shown that CGRP has some affinity for the CLR/RAMP3 complex (AM2 receptor) [24].

### 2.4. Intracellular Signaling

The binding of CGRP to the CGRP1 receptor results in increased intracellular cAMP levels, activation of protein kinase A, and the opening of ATP-sensitive K^+^ channels [26]. This pathway is primarily responsible for smooth muscle cell relaxation and vasodilation. Another pathway that leads to endothelium-dependent vasodilation is associated with the stimulation of nitric oxygen (NO) release [27].

Activation of protein kinase A also induces the expression of the c-fos gene, which appears to be involved in pain sensitization [28]. CGRP can activate mitogen-activated protein kinases (MAPK) that promote fibroblast proliferation [29]. Furthermore, it can stimulate antiapoptotic pathways involving MAPK, ERK1/2, and p38 [30]. Activation of ERK and p38 also appears to modulate the response to morphine [31].

## 3. Physiological and Pathophysiological Effect of CGRP

### 3.1. CGRP in the Cardiovascular System

#### 3.1.1. CGRP as a Vasodilator

CGRP can induce vasodilation and increase blood flow in various organs. For example, the injection of femtomoles of CGRP causes an increase in microcirculation in the skin, brain, coronary, and renal regions [10,32]. As previously discussed, the vasodilation mechanism of CGRP involves both the NO-dependent pathway and the NO-independent pathway [33].

#### 3.1.2. CGRP in the Regulation of Systemic Circulation

CGRP can influence blood pressure regulation. It is believed that CGRP is released in response to hypertensive stimuli such as angiotensin II activation and postural changes [34] and plays an important role in modulating baroreceptor reflex sensitivity [35]. Despite CGRP administration resulting in a reduction in systemic arterial pressure [13,33,36], it is important to note that it does not act as a hypotensive mediator since it simultaneously causes a positive inotropic and chronotropic response in the heart, preventing hypotension [37,38]. The function of CGRP, therefore, appears to be that of a regulator and modulator of the pressor response. This hypothesis is supported by the fact that CGRP antagonists do not seem to cause a significant pressor effect, likely due to the consequent activation of vasodilatory mediators such as NO and prostaglandins [39,40]. Furthermore, studies on genetic polymorphisms in the CALC1 gene have shown an association between specific genetic variants, such as T-692C, and an increased susceptibility to the development of arterial hypertension [41].

#### 3.1.3. CGRP Physiologic Effect in the Heart

CGRP plays a cardioprotective role [1]. It is found in fibers innervating coronary arteries, papillary muscle, and the sinoatrial and atrioventricular nodes [42]. As previously mentioned, CGRP has a positive effect on cardiac chronotropy and inotropy [43]. Furthermore, CGRP promotes coronary vasodilation, even if its effect is predominant in the distal portion of the coronary vessels [44], and is involved in the formation of new blood vessels in response to ischemic insults [45]. Other beneficial effects of CGRP include reducing inflammation and macrophage accumulation, decreasing apoptosis, and post-infarction fibrosis [46]. Indeed, CGRP plays an antiapoptotic role in cardiomyocytes by activating the ERK1/2 signaling pathway [47].

#### 3.1.4. CGRP in Myocardial Infarction and Heart Failure

Following myocardial infarction or during acute heart failure, increased levels of CGRP are observed as a response to metabolic stress and reduced coronary vasodilation [48,49]. Endogenous CGRP administration improves coronary circulation in patients with cardiac pathologies [50,51,52] and performance during exercise tests [53]. CGRP also plays an important role in cardiac preconditioning, providing greater resistance of cardiomyocytes to apoptosis and reducing the inflammatory response [54]. It has also been shown that individuals with coronary artery disease have lower levels of CGRP compared to healthy subjects [55]. Importantly, CGRP inhibitors and antagonists do not seem to increase the risk of ischemia in healthy individuals [56].

#### 3.1.5. CGRP in Atherosclerosis and Vascular Remodeling

Atherosclerosis is characterized by endothelial dysfunction, reduced NO production, intimal inflammation, and the formation of atherosclerotic plaques [57]. CGRP plays a protective role on the endothelium by reducing inflammation and promoting neointimal formation [58,59,60]. Nevertheless, monoclonal antibodies targeting the CGRP receptor (i.e., erenumab) did not modify flow-mediated (endothelial-dependent) dilation in treated patients [61]. Furthermore, CGRP reduces vascular remodeling through increased cAMP levels [62].

### 3.2. CGRP in the Skin

CGRP plays various roles in the skin. Physiologically, CGRP can be released not only by sensory afferents but also by keratinocytes and immune cells [63]. Primarily, it causes cutaneous vasodilation, increasing blood flow to the affected region in response to various stimuli [32]. For example, it is a modulator of the “skin axon reflex flare” phenomenon, which causes cutaneous vasodilation after stimulation of C-fiber nerve roots and, together with antidromic vasodilation, is responsible for flushing [64]. Increased plasma and urinary levels of CGRP have been observed in patients with postmenopausal flushing [65]. Another function appears to be thermoregulation through the regulation of vasodilation in dermal arterioles [1]. The vasomotor effects of CGRP lead to increased cutaneous edema formation in response to contusions and wounds, with increased vessel permeability and leukocyte accumulation in the skin [64,66]. It also plays a role in immune regulation in atopic dermatitis, reducing the inflammatory response [67].

#### CGRP in Cutaneous Wound Healing

CGRP enhances wound healing and modulates pain perception [68]. In CGRP gene knockout mice, a reduction in tissue repair is observed [69]. This effect may be associated with CGRP’s ability to stimulate vascular endothelial growth factor (VEGF), promoting neovascularization in the injured area. Furthermore, CGRP appears to promote the proliferation of keratinocytes [70] and fibroblasts [71].

### 3.3. CGRP in the Respiratory System

CGRP plays various roles in the respiratory system. It is synthesized by pulmonary neuroendocrine cells (PNECs) and is co-localized with substance P in the sensory C fibers [72]. It actively contributes to pulmonary homeostasis by causing vasodilation, and bronchoprotection, modulating the inflammatory response [73] and promoting proliferation of bronchial epithelial cells [74].

#### 3.3.1. CGRP and Asthma

In the context of asthma, CGRP plays a complex role, having both pro-inflammatory and anti-inflammatory effects. On one hand, it activates the Th9 response and induces IL-9 and leukotriene C4 synthesis, leading to bronchial edema [1,73]. On the other hand, it suppresses eosinophilic inflammatory response, similar to anti-IL-5 monoclonal antibodies [73]. It also inhibits the activation of Th2 lymphocytes, suppressing the synthesis of type 2 cytokines, and stimulates the proliferation of regulatory T lymphocytes [75]. CGRP has bronchodilatory effects, improving lung ventilation [73]. Studies are underway to evaluate the effectiveness of a CGRP agonist in bronchial asthma [73].

#### 3.3.2. CGRP and Pulmonary Hypertension

The significant regulatory effect of CGRP on vascular tone also affects the development of pulmonary hypertension. CGRP appears to have a protective role in the development of this condition, as evidenced by reduced plasma levels in rats with pulmonary hypertension [76]. This effect may be secondary to the suppression of endothelin-1 (ET-1) and angiotensin II (ANG2), contributing to the reduction in pulmonary resistance [77]. It has also been suggested that the degeneration of capsaicin-sensitive sensory nerve fibers due to the loss of CGRP may contribute to increased pulmonary pressure in rats [78]. The supplementation of CGRP has been proposed for the treatment of pulmonary hypertension [79].

#### 3.3.3. CGRP and COVID-19

The FDA has approved the clinical use of CGRP antagonists to reduce lung inflammation and acute respiratory distress syndrome (ARDS) in COVID-19 patients [75]. It is believed that CGRP may stimulate IL-6 and polarize the immune response toward Th17 lymphocytes, which represent one of the main pathogenic mechanisms of COVID-19 [75]. However, there are concerns about the effectiveness and safety of these drugs due to the potential beneficial modulatory role of CGRP in pulmonary inflammation [75].

### 3.4. CGRP and the Gastrointestinal System

CGRP plays various functions in the gastrointestinal tract. In the gastrointestinal tract, two main autonomic nerve afferents are present: the parasympathetic component (vagus and pelvic nerves) and the sympathetic component (splanchnic and hypogastric nerves) [80]. Vagal afferents terminate in the myenteric plexus, circular muscle, and submucosal plexus, while sympathetic afferents are found in the myenteric plexus, submucosa, and mucosa surrounding blood vessels [80]. CGRP and substance P are found in the enteric plexus, interganglionic fibers, muscular layer, and mucosa of both the stomach and intestine. CGRP seems to regulate motility, inflammation, gastric acid secretion, and inflammation in the gastrointestinal system [80].

#### 3.4.1. CGRP and the Stomach

CGRP plays a fundamental role in the protection of gastric mucosa [80]. It appears to inhibit gastric acid secretion and gastric motility [81]. Release from the dorsal roots of the ganglia and vagal ganglia of splanchnic nerves also leads to increased blood flow in the gastric mucosa [80]. It exhibits anti-inflammatory and anti-apoptotic effects on gastric mucosal cells damaged by ischemic processes [80]. CGRP antagonists seem to increase gastric lesions mediated by indomethacin and ethanol, as demonstrated in studies conducted on CGRP knockout mice [82]. The release of CGRP through the activation of TRPV-Q has been shown to improve epigastric pain in patients with dyspepsia and irritable bowel syndrome [83]. Furthermore, the TRPV1 receptors that stimulate CGRP release have been analyzed as potential targets to prevent gastric mucosal damage [84]. The effect of CGRP on gastric motility seems to be mediated both by stimulation of CGRP 1 reception and by activation of amylin receptor 1. CGRP inhibits gastrointestinal transit and gastric emptying [85]. Antagonists of CGRP such as triptans, gepant, and anti-CGRP antibodies are known to lead to gastrointestinal symptoms such as stomach pain, nausea, and vomiting [86,87]. CGRP seems to interact also with Glucagon-Like Peptide-1 (GLP-1) increasing its values by >60% with a bidirectional regulation as GLP-1 has been shown to induce secretion of CGRP [88,89]. Anti-CGRP drugs seem indeed to reduce GLP-1 secretion [88].

#### 3.4.2. CGRP and Metabolic Syndrome

CGRP is released in the pancreas, leading to a reduction in insulin release and an increase in blood glucose levels [90], as well as insulin resistance in muscle. The use of capsaicin to destroy CGRP-containing sensory nerves results in increased glucose tolerance [91]. It also appears to contribute to weight gain, as evidenced by a study on αCGRP knockout mice, which showed a reduction in diet-induced obesity incidence [92]. The activity of sensory nerves releasing CGRP appears to be increased in obesity and metabolic syndrome [1]. Conversely, CGRP has been shown to have a role in appetite reduction. One hour after a meal, an increase in CGRP release in the central nervous system and intestine is observed, with functions similar to insulin [93].

### 3.5. CGRP and the Musculoskeletal System

Elevated levels of CGRP have been found in the synovial fluid of patients with arthritis [12,94,95,96], suggesting that it may be considered an early mediator of the disease. CGRP can induce cytokine production by leukocytes and fibroblasts in rheumatoid arthritis and osteoarthritis [95,97]. CGRP antagonists inhibit synovial cell proliferation and the production of pro-inflammatory cytokines [98]. Furthermore, CGRP 8-37, a CGRP antagonist, appears to inhibit the hypersensitivity of joint nerve endings responsible for arthritic pain [99]. Interestingly, methotrexate, a drug used in the treatment of rheumatoid arthritis, reduces the presence of CGRP-positive fibers [100]. Although CGRP appears to play a predominant role in inflammatory arthritis, elevated CGRP expression has also been observed in neurons of joints affected by osteoarthritis [94]. Additionally, the drug LY2951742, which neutralizes CGRP, appears to have a protective role in osteoarthritis in mice [101]. Finally, CGRP also seems to play a role also on bone homeostasis and regeneration [102]. The Transient receptor potential vanilloid type 1 (TRPV1) activation in the dorsal root ganglion contributes essentially to the regulation of osteoblast physiology through the modulation of the production and secretion of CGRP [103].

### 3.6. CGRP and the Urinary System

CGRP is present in renal fibers, primarily in the muscle layer of the pelvis, as well as in peri-glomerular and peritubular arteries and arterioles [104]. It is also found in the ureters and bladder. At the glomerular level, it causes vasodilation and an increase in renal blood flow, leading to a reduction in mean renal pressure [105] and an increase in glomerular filtration [106]. At the tubular level, it increases the excretion of sodium, chloride, potassium, calcium, and phosphates, stimulating urine flow [107,108]. It also appears to increase renin release, although it is unclear whether it acts directly or through its pressor effects on the glomerulus [104].

### 3.7. CGRP and Sex Hormones

The vasodilatory properties of CGRP are exerted also during pregnancy. The circulating CGRP levels increase during gestation while declining rapidly at term and in the postpartum. In addition, in rats, the sensitivity of various vascular beds to CGRP increases with advancing pregnancy. This action may be essential in regulating uteroplacental blood flow, in addition to other vascular adaptations. Furthermore, the activation of the CGRP pathway is associated with uterine relaxation, possibly as the consequence of the upregulation of CGRP receptors. In this light, sex steroid hormones, estrogens, and progesterone regulate CGRP synthesis and its effects on both myometrial and uterine vascular tissues [109]. CGRP also possibly interacts with oxytocin, another relevant hormone with anti-nociceptive properties also relevant to uterine tone [110]. The systemic effects of CGRP are synthesized in Table 1.

## 4. Discussion

The calcitonin gene-related peptide (CGRP) is a molecule that exhibits considerable complexity in its biological activities, displaying effects in various body systems that may, in some cases, appear contradictory. It serves as a molecule that regulates several processes both directly and indirectly, playing a primarily homeostatic role [1]. This characteristic may underlie the low incidence of side effects observed with drugs that interact with CGRP, such as monoclonal antibodies and gepants [87]. The main side effects of these drugs are predominantly gastrointestinal, where CGRP plays a prominent role, slowing gastric emptying, interacting with GLP-1, and reducing hydrochloric acid secretion [81,88]. Both gepants and anti-CGRP monoclonal antibodies have shown common side effects of constipation and nausea [87,111].

Although the main effect of CGRP is to modulate vascular smooth muscle tone, leading to vasodilation, the incidence of such side effects in drugs that antagonize CGRP’s effects is minimal [112], as observed in real-life studies [113,114]. As previously mentioned, the use of CGRP receptor antagonists in mice does not lead to vasoconstriction and arterial hypertension because there is a concomitant release of molecules such as NO and prostaglandins, which ultimately induce relaxation of smooth muscle [39,40]. This review highlights that CGRP’s effect on the vascular system is not exclusively that of a direct vasodilator, but rather that of a regulator that reduces susceptibility to the development of chronic arterial hypertension, as demonstrated by studies on CALC-1 gene polymorphisms [41]. Similarly, at the cardiac level, CGRP has shown protective roles toward cardiomyocytes, reducing apoptosis and inflammation and leading to improved coronary circulation [46,47]. Nonetheless, few ischemic side effects have been observed with therapies with CGRP inhibitors [112]. Studies have demonstrated improved exercise performance after CGRP infusion, but, on the other hand, a randomized trial did not show differences in terms of exercise performance during treadmill testing in patients with stable angina undergoing treatment with anti-CGRP antibodies versus placebo [115]. In the cardiovascular context, it would be beneficial to analyze the long-term effects of these drugs.

The immune role of CGRP is extremely complex and varied. It appears to have both anti-inflammatory and anti-apoptotic roles, as well as pro-inflammatory effects. In the lungs, it directs the immune response toward Th9 lymphocytes while reducing the Th2 response, resulting in increased bronchial edema and, conversely, suppressing eosinophils and causing bronchodilation. At the cutaneous level, vasodilation mediated by CGRP leads to increased edema and passage of leukocytes from the circulatory system to the dermis. On the other hand, CGRP accelerates wound healing processes. It appears to have an anti-inflammatory role in arthritic and arthrosis conditions. A case series has shown the presence of inflammatory complications in patients receiving CGRP inhibitors, such as autoimmune hepatitis, Susac syndrome, DRESS syndrome, Weber syndrome, severe polyarthralgia, exacerbation of psoriasis, and urticarial eczema, after 1–16 months of treatment [11].

## 5. Conclusions

CGRP represents a regulator of diverse biological processes, including nociception, vasodilation, metabolism, inflammation, and gastrointestinal emptying (Figure 1).

Many of the effects it exerts also appear to have long-term implications. These aspects become even more relevant, considering that migraine, the present indication for CGRP pathway targeted therapies, is often comorbid with disorders involving the same systems [115,116]. Therefore, it is crucial to continue investigating CGRP activities and to be mindful of potential effects resulting from chronic use of its inhibitors especially if comorbid conditions coexist. In this light, to gain a holistic and cutting-edge perspective on CGRP’s functions and its potential applications in diverse somatic domains, future approaches may also include quantitative systems pharmacology modeling, multi-omics analyses, advanced neuronal manipulation techniques, 3D organoid models, machine learning predictive modeling, and synthetic biology [117].

## Figures and Tables

**Figure 1 ijms-24-13979-f001:**
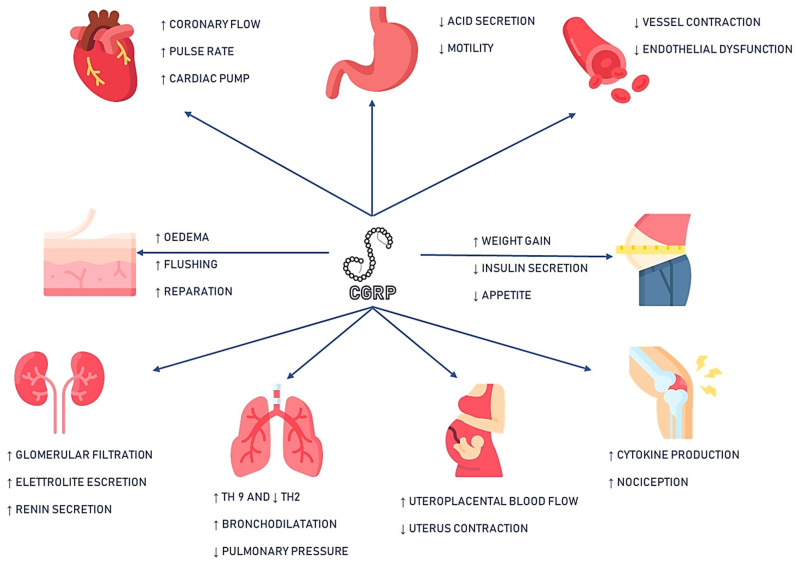
Synthesis of CGRP systemic effects. Icon’s copyright is to Flaticon (Freepik, Futuer, Vitaly Gorbachev, Smashicon, MindWorlds, England, UK) https://www.flaticon.com/ (accessed on 30 June 2023) Legend: ↑: increase ↓: decrease.

**Table 1 ijms-24-13979-t001:** CGRP and its antagonists’ systemic effects.

District	Function	Antagonists’ Effect
Arterial pressure	Vasodilatation in response to hypertensive stimuli (e.g., angiotensin II, postural changes) [34]. Protection against hypertension development [41].	No significant effect on pressure [39,40].
Heart	Chronotropic and inotropic positive response [37,38]. Anti-inflammatory, anti-apoptotic effects [47,48,49]. Coronary vasodilatation [44].	No significant increase in myocardial infarction incidence [56]
Vessel	Reduction in endothelial inflammation and vascular remodeling [58,59,60,62].	Not known
Skin	Induces flushing [64]. Increases edema formation [64,66] and stimulates tissue reparation in response to contusion and wound [69]. Reduction in inflammation in atopic dermatitis [67].	Not known
Lung	Activates Th9 response [1,73]. Reduces Th2 and eosinophilic response [75]. Induces bronchodilatation [73]. Induces bronchial edema [1,73]. Protection from pulmonary hypertension through suppression of endothelin-1 and angiotensin II [77].	Not known. Under study anti-CGRP drugs for COVID-19 [75].
Gastrointestinal tract	Inhibition of gastric acid secretion, gastric, and intestinal motility [80,81]. Reduce inflammation and apoptosis in ischemic and alcohol-mediated stomach lesions [80,82].	Can cause nausea, constipation, or diarrhea and vomiting [86,87]. Seems to reduce GLP-1 CGRP-induced secretion [88].
Metabolism	Reduce insulin release after meal [90]. Contributes to weight gain [92]. Reduces appetite [92].	Reduction in diet-induced obesity incidence [92].
Joint	Induces cytokine production in rheumatoid arthritis and osteoarthritis [95,97].	Inhibits cell proliferation and production of pro-inflammatory cytokines [98]. Reduces hypersensitivity of joint nerves [99].
Kidney	Vasodilatation and increase in renal blood flow and glomerular filtration [105,106]. Increasing excretion of electrolytes [107,108]. Increase renin release [104].	Not known
Pregnancy	Regulation of uteroplacental blood flow [109]. Relaxation of uterus. Interaction with oxytocin [110].	Not known

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
