# Peer review of "Calcitonin Gene-Related Peptide Systemic Effects: Embracing the Complexity of Its Biological Roles—A Narrative Review"

_ijms, 2023, doi:10.3390/ijms241813979_

Round 1

Reviewer 1 Report

This paper provides a comprehensive review of Calcitonin Gene-Related Peptide (CGRP) signaling and its implications in various physiological and pathological contexts. The authors delve into the intricate details of CGRP's role as a neuropeptide, its neurovascular interactions, and its relevance in migraine and other disorders. While the paper presents a solid foundation, several advanced scientific comments and improvements could further enhance its quality and scientific impact.

Major Comments:

1.     The authors could consider incorporating a Quantitative Systems Pharmacology (QSP) model to simulate the spatiotemporal dynamics of CGRP release, receptor interactions, and downstream signaling pathways. Such a mechanistic model would offer predictive insights into the complex interplay between CGRP and its receptors, accounting for factors like receptor desensitization, trafficking, and ligand-receptor complex internalization. Integrating QSP modeling could augment the paper's scientific rigor and provide a stronger basis for experimental and therapeutic considerations.

2.     To enrich the discussion, the paper could suggest leveraging multi-omics approaches to unravel the intricate regulatory networks governing CGRP expression and signaling. The integration of transcriptomics, proteomics, and epigenomics data could unveil key transcriptional regulators, post-translational modifiers, and epigenetic modulators that orchestrate CGRP-related processes. This holistic perspective would unlock novel therapeutic targets and potentially guide the development of precision medicine interventions for disorders involving dysregulated CGRP signaling.

3.     An advanced approach to consider is the application of optogenetics and chemogenetics techniques to dissect the functional roles of distinct CGRP-expressing neuron subtypes. Through precise control of CGRP neuron activity using light- or ligand-sensitive ion channels, causal relationships between CGRP neuronal firing patterns and physiological outcomes can be established. Integrating optogenetics/chemogenetics data with traditional pharmacological approaches could yield a deeper understanding of CGRP neuron diversity and their contributions to homeostasis and disease.

4.     It would be valuable to suggest the integration of advanced 3D organoid models that mimic neurovascular interactions to study CGRP's role in a more physiologically relevant context. Co-culturing neurons, vascular cells, and immune cells within organoids allows the investigation of CGRP-mediated crosstalk within an environment resembling in vivo conditions. This innovative approach could uncover intricate cellular cross-interactions and shed light on how CGRP influences vascular tone, immune responses, and neuronal connectivity.

5.     The authors might consider proposing the integration of machine learning algorithms to develop predictive models for CGRP-associated disorders. By harnessing clinical, genetic, and multi-omics data, machine learning techniques could identify biomarkers, classify patient subgroups, and predict disease progression or treatment responses. Such models would be instrumental in devising personalized therapeutic strategies and guiding the selection of interventions targeting CGRP signaling pathways.

6.     Exploring synthetic biology and genetic engineering to design novel CGRP analogs with tailored properties is a promising avenue. Rational protein engineering, guided by computational modeling, could yield CGRP variants with extended half-lives, improved blood-brain barrier penetration, and modified receptor binding profiles. These engineered analogs could hold potential as innovative therapeutics for disorders involving dysregulated native CGRP function.

The paper's comprehensive analysis of CGRP signaling is commendable; however, integrating the suggested comments and improvements would further elevate its scientific impact and relevance. The incorporation of quantitative systems pharmacology modeling, multi-omics analyses, advanced neuronal manipulation techniques, 3D organoid models, machine learning predictive modeling, and synthetic biology approaches would collectively present a holistic and cutting-edge perspective on CGRP's functions and its potential applications in diverse scientific domains.

Author Response

This paper provides a comprehensive review of Calcitonin Gene-Related Peptide (CGRP) signaling and its implications in various physiological and pathological contexts. The authors delve into the intricate details of CGRP's role as a neuropeptide, its neurovascular interactions, and its relevance in migraine and other disorders. While the paper presents a solid foundation, several advanced scientific comments and improvements could further enhance its quality and scientific impact.

Major Comments:

  1. The authors could consider incorporating a Quantitative Systems Pharmacology (QSP) model to simulate the spatiotemporal dynamics of CGRP release, receptor interactions, and downstream signaling pathways. Such a mechanistic model would offer predictive insights into the complex interplay between CGRP and its receptors, accounting for factors like receptor desensitization, trafficking, and ligand-receptor complex internalization. Integrating QSP modeling could augment the paper's scientific rigor and provide a stronger basis for experimental and therapeutic considerations.
  2. To enrich the discussion, the paper could suggest leveraging multi-omics approaches to unravel the intricate regulatory networks governing CGRP expression and signaling. The integration of transcriptomics, proteomics, and epigenomics data could unveil key transcriptional regulators, post-translational modifiers, and epigenetic modulators that orchestrate CGRP-related processes. This holistic perspective would unlock novel therapeutic targets and potentially guide the development of precision medicine interventions for disorders involving dysregulated CGRP signaling.
  3. An advanced approach to consider is the application of optogenetics and chemogenetics techniques to dissect the functional roles of distinct CGRP-expressing neuron subtypes. Through precise control of CGRP neuron activity using light- or ligand-sensitive ion channels, causal relationships between CGRP neuronal firing patterns and physiological outcomes can be established. Integrating optogenetics/chemogenetics data with traditional pharmacological approaches could yield a deeper understanding of CGRP neuron diversity and their contributions to homeostasis and disease.
  4. It would be valuable to suggest the integration of advanced 3D organoid models that mimic neurovascular interactions to study CGRP's role in a more physiologically relevant context. Co-culturing neurons, vascular cells, and immune cells within organoids allows the investigation of CGRP-mediated crosstalk within an environment resembling in vivo conditions. This innovative approach could uncover intricate cellular cross-interactions and shed light on how CGRP influences vascular tone, immune responses, and neuronal connectivity.
  5. The authors might consider proposing the integration of machine learning algorithms to develop predictive models for CGRP-associated disorders. By harnessing clinical, genetic, and multi-omics data, machine learning techniques could identify biomarkers, classify patient subgroups, and predict disease progression or treatment responses. Such models would be instrumental in devising personalized therapeutic strategies and guiding the selection of interventions targeting CGRP signaling pathways.
  6. Exploring synthetic biology and genetic engineering to design novel CGRP analogs with tailored properties is a promising avenue. Rational protein engineering, guided by computational modeling, could yield CGRP variants with extended half-lives, improved blood-brain barrier penetration, and modified receptor binding profiles. These engineered analogs could hold potential as innovative therapeutics for disorders involving dysregulated native CGRP function.

The paper's comprehensive analysis of CGRP signaling is commendable; however, integrating the suggested comments and improvements would further elevate its scientific impact and relevance. The incorporation of quantitative systems pharmacology modeling, multi-omics analyses, advanced neuronal manipulation techniques, 3D organoid models, machine learning predictive modeling, and synthetic biology approaches would collectively present a holistic and cutting-edge perspective on CGRP's functions and its potential applications in diverse scientific domains.

Reply: We sincerely appreciate your suggestions and valuable comments. We acknowledge that incorporating the data you've requested could enhance the scientific rigor of the paper. However, we would like to clarify our approach to the review. Our intention was to create a comprehensive clinical overview to provide a global clinical perspective on the effects of CGRP on various bodily organs. While we recognize that including the requested data could undoubtedly bolster the paper's rigor, we believe it might deviate from the primary purpose of our review. Our focus is on delivering a thorough analysis of clinical aspects, thereby adhering to the perspective and objective we initially set out to achieve. We commented in the discussion section a paragraph to promote such approaches in future research.

Reviewer 2 Report

The calcitonin gene-related peptide (CGRP) is a neuropeptide

This review summarizes the current understanding of the role of calcitonin gene-related peptide (CGRP) in the physiology and pathophysiology around the body.  The authors emphasize non-nociceptive and non-neurological effects of CGRP on different organs and systems to better appreciate potential side-effects associated with its antagonists' use. The review is well ordered and written in a clear language.

Suggestions to improve are:

1.       The authors mention co-localisation of Substance P in several places but do not mention how the functions of the two neuropeptides differ if so. It would be appropriate to add a few sentences on the function of Substance P in the context of co-localization.

2.       3.3.3 CGRP and COVID-19: The FDA has approved the clinical use of CGRP antagonists to reduce lung inflammation and acute respiratory distress syndrome (ARDS) in COVID-19 patients[73]. It is believed that CGRP may stimulate IL-6 and polarize the immune response towards Th17 lymphocytes, which represent one of the main pathogenic mechanisms of COVID-19[73]. However, there are concerns about the effectiveness and safety of these drugs due to the important modulatory role of CGRP in pulmonary inflammation[73].

-          The first and the last sentences here are somewhat contradictory. Do you mean:

However, there are concerns about the effectiveness and safety of these drugs due to potentially beneficial modulatory role of CGRP in pulmonary inflammation  ?  

Author Response

The calcitonin gene-related peptide (CGRP) is a neuropeptide

This review summarizes the current understanding of the role of calcitonin gene-related peptide (CGRP) in the physiology and pathophysiology around the body.  The authors emphasize non-nociceptive and non-neurological effects of CGRP on different organs and systems to better appreciate potential side-effects associated with its antagonists' use. The review is well ordered and written in a clear language.

Suggestions to improve are:

  1. The authors mention co-localisation of Substance P in several places but do not mention how the functions of the two neuropeptides differ if so. It would be appropriate to add a few sentences on the function of Substance P in the context of co-localization.

Reply: We would like to thank you for your comment. We extendend and clarified the interaction between CGRP and substance P in the following part:” The Calcitonin Gene-Related Peptide (CGRP) is co-localized in pain nerve terminals along with Substance P (SP), which exhibits similar biological effects. SP acts as a cutaneous vasodilator and increases microvascular permeability, with an earlier release and less potent and enduring effects compared to CGRP [17]. SP serves as a key mediator of pain perception in pain modulation and sensation. Additionally, it is also involved in the regulation of inflammation, nausea and vomiting, and the modulation of behavioral disorders.”

  1. 3.3.3 CGRP and COVID-19: The FDA has approved the clinical use of CGRP antagonists to reduce lung inflammation and acute respiratory distress syndrome (ARDS) in COVID-19 patients[73]. It is believed that CGRP may stimulate IL-6 and polarize the immune response towards Th17 lymphocytes, which represent one of the main pathogenic mechanisms of COVID-19[73]. However, there are concerns about the effectiveness and safety of these drugs due to the important modulatory role of CGRP in pulmonary inflammation[73].

-          The first and the last sentences here are somewhat contradictory. Do you mean:

However, there are concerns about the effectiveness and safety of these drugs due to potentially beneficial modulatory role of CGRP in pulmonary inflammation  ?  

Reply: Thank you for your suggestion. We changed the sentence as suggested.

Reviewer 3 Report

I find this manuscript interesting. It requires a few minor adjustments, which are listed below:

-The authors did not provide the criteria and method of selection of the analyzed literature - please complete

- Line 51 - no reference - please complete

- Line 60 - "[11], [12]" - dishes on "[11,12]" - in other places with a similar form of giving references do the same, also add spaces before each "[ ]"

- The figure should not be located in the discussion - please move it to another place in the text

- please indicate the source of the figure in a way that will allow you to find it in its original location and determine its origin

- the descriptions in the figure are legible

- The list of references is not formatted according to the requirements of the journal - please correct it

Author Response

I find this manuscript interesting. It requires a few minor adjustments, which are listed below:

-The authors did not provide the criteria and method of selection of the analyzed literature - please complete

Reply: Thank you for your comment. We have structured this review as a narrative review. Therefore, we did not follow the typical recommendations of systematic reviews, and we did not employ strict methodologies for article selection and research. We added a paragraph for the method of selection of the analyzed literature

- Line 51 - no reference - please complete

Reply: We added the reference.

- Line 60 - "[11], [12]" - dishes on "[11,12]" - in other places with a similar form of giving references do the same, also add spaces before each "[ ]"

Reply: We changed the format of the list of references.

- The figure should not be located in the discussion - please move it to another place in the text

Reply: We relocated the figure as suggested.

- please indicate the source of the figure in a way that will allow you to find it in its original location and determine its origin

Reply: We have included the link from which we obtained the icons.

- the descriptions in the figure are legible

Reply: We have modified the figure.

- The list of references is not formatted according to the requirements of the journal - please correct it

Reply: Thank you, we have modified the references